# Extracellular Hsp70 and Circulating Endometriotic Cells as Novel Biomarkers for Endometriosis

**DOI:** 10.3390/ijms252111643

**Published:** 2024-10-30

**Authors:** Christiane Guder, Soraya Heinrich, Vanadin Seifert-Klauss, Marion Kiechle, Lisa Bauer, Rupert Öllinger, Andreas Pichlmair, Marie-Nicole Theodoraki, Veena Ramesh, Ali Bashiri Dezfouli, Barbara Wollenberg, Alan Graham Pockley, Gabriele Multhoff

**Affiliations:** 1Department of Otholaryngology, Head and Neck Surgery, TUM University Hospital, School of Medicine and Health, Technical University Munich, Ismaningerstr. 21, 81675 Munich, Germany; christiane.guder@tum.de (C.G.); marie-nicole.theodoraki@uniklinik-ulm.de (M.-N.T.); veena.ramesh@tum.de (V.R.); ali.bashiri@tum.de (A.B.D.); barbara.wollenberg@tum.de (B.W.); 2Department of Gynecology and Obstetrics, TUM University Hospital, School of Medicine and Health, Technical University Munich, Ismaningerstr. 21, 81675 Munich, Germany; soraya.heinrich@tum.de (S.H.); vanadin.seifert-klauss@tum.de (V.S.-K.); marion.kiechle@tum.de (M.K.); 3Radiation Immuno-Oncology, TranslaTUM—Central Institute for Translational Cancer Research, TUM University Hospital, School of Medicine and Health, Technical University Munich, Ismaningerstr. 21, 81675 Munich, Germany; lisa.bauer@tum.de; 4Department of Radiation Oncology, TUM University Hospital, School of Medicine and Health, Technical University Munich, Ismaningerstr. 21, 81675 Munich, Germany; 5Institute of Molecular Oncology and Functional Genomics, TranslaTUM—Central Institute for Translational Cancer Research, TUM University Hospital, School of Medicine and Health, Technical University Munich, Ismaningerstr. 21, 81675 Munich, Germany; rupert.oellinger@tum.de; 6Department of Virology, TranslaTUM—Central Institute for Translational Cancer Research, TUM University Hospital, School of Medicine and Health, Technical University Munich, Ismaningerstr. 21, 81675 Munich, Germany; andreas.pichlmair@tum.de; 7Department of ENT, Head and Neck Surgery, University Hospital Ulm, Albert Einstein-Allee 23, 89070 Ulm, Germany; 8John van Geest Cancer Research Centre, School of Science and Technology, Nottingham Trent University, Nottingham NG11 8NS, UK; graham.pockley@ntu.ac.uk

**Keywords:** endometriosis, mesenchymal stemness biomarker, heat shock protein 70 (Hsp70)

## Abstract

Stress-inducible heat shock protein 70 (Hsp70), which functions as a molecular chaperone and is frequently overexpressed in different cancer cell types, is present on the cell surface of tumor cells and is actively released into the circulation in free and extracellular lipid vesicle-associated forms. Since the exact pathomechanism of endometriosis has not yet been elucidated (although it has been associated with the development of endometrial and ovarian cancer), we asked whether extracellular Hsp70 and circulating endometriotic cells (CECs) reflect the presence and development of endometriosis. Therefore, circulating levels of free and lipid microvesicle-associated Hsp70 were measured using the Hsp70-exo ELISA, and the presence of circulating CECs in the peripheral blood of patients with endometriosis was determined using membrane Hsp70 (mHsp70) and EpCAM monoclonal antibody (mAb)-based bead isolation approaches. Isolated CECs were further characterized by immunofluorescence using reagents directed against cytokeratin (epithelial marker), CD45 (leukocyte marker), CD105/CD44 (mesenchymal stemness markers) and by comparative RNA analysis. Similar to the situation in patients with cancer, the levels of circulating Hsp70 were elevated in the blood of patients with histologically proven endometriosis compared to a healthy control cohort, with significantly elevated Hsp70 levels in endometriosis patients with lesions outside the uterine cavity. Moreover, CECs could be isolated using the cmHsp70.1 mAb-based, and to a lesser extent EpCAM mAb-based, bead approach in all patients with endometriosis, with the highest counts obtained using the mHsp70-targeting procedure in patients with extra-uterine involvement. The longevity in cell culture and the expression of the cytokeratins CD105 and CD44, together with differentially expressed genes related to epithelial-to-mesenchymal transition (EMT), revealed similarities between mHsp70-expressing CECs and circulating tumor cells (CTCs) and suggest a mesenchymal stem cell origin. These findings support the involvement of mHsp70-positive stem cell-like cells in the development of endometriotic lesions. In summary, elevated levels of Hsp70 and CECs in the circulation could serve as liquid biopsy markers for endometriosis with extra-uterine involvement and help to elucidate the underlying pathomechanism of the disease.

## 1. Introduction

Worldwide, 10–15% of all women are affected by endometriosis, an estrogen-dependent benign gynecological disease [1,2]. The disease is characterized by the presence of functional endometrial cells from outside of the endometrium, predominantly from ovarian and other pelvic tissues, but also in more distant organs such as the brain, lung and lymph nodes [1,2]. Due to the non-specific symptoms of endometriosis, such as pain during menses, chronic pelvic pain and migraines, the diagnosis of endometriosis is often delayed by up to 10 years. Presently, only invasive laparoscopic methods can diagnose endometriosis using biopsies [3]. Therefore, there is a high unmet clinical need for reliable liquid biopsy-based biomarkers that can improve the early diagnosis of endometriosis and can be repeatedly analyzed by minimally invasive methods.

In addition to retrograde menstruation, immunopathological and endocrine factors as well as stem cells have been suggested to play a role in the development of endometriosis [2,4]; however, the exact pathomechanism has not yet been elucidated. Since endometriotic cells are not locally restricted to the endometrium, and can migrate into extra-genital organs (most likely via the bloodstream), we investigated the blood as a source of potential biomarkers for endometriosis.

Members of the 70 kDa heat shock (stress) protein family (HSP70) reside in nearly all subcellular compartments such as the cytosol, nucleus, lysosomes, endoplasmic reticulum (ER) and mitochondria. The highly conserved, major stress-inducible Hsp70, a member of this HSP family, is responsible for maintaining protein homeostasis, protein transport, antigen processing and the prevention of cell death [5]. High intracellular expression of Hsp70 in tumor cells is associated with tumor progression, anti-apoptotic signaling and therapy resistance [6,7]. In tumor cells, but not normal cells, Hsp70 can be translocated from the cytosol to the plasma membrane by an ER-independent, non-classical, vesicular transport mechanism [8,9]. It is well established that tumor cells overexpressing Hsp70 in the cytosol [6] and membrane Hsp70 (mHsp70) on the cell surface [10] have the capacity to release Hsp70 into extracellular lipid microvesicles with biophysical properties of exosomes [11]. Previous data from our group demonstrated that elevated levels of Hsp70 in the blood can serve as a biomarker for a large variety of different tumor entities such as breast, lung, head and neck, and brain cancers [12,13,14,15,16]. In this study, we analyzed Hsp70 levels in the circulation of patients with endometriosis and patients with endometriosis and extra-uterine involvement in the bladder, rectum and lymph nodes to evaluate its role as a potential biomarker of disease. 

Circulating tumor cells (CTCs) are the most commonly isolated cells using antibody-based methods targeting the expression of epithelial adhesion molecule (EpCAM, CD326), which is crucial for the malignancy of epithelial tumors [17,18]. After entering the bloodstream, CTCs migrating to different organs in the body extravasate into a different microenvironment. By evading the innate and adaptive immune responses in the blood and being able to survive independently of other cells as a cell clone or colony, and by adapting to a new microenvironment, proliferating and forming vessels, CTCs fulfil certain criteria that are essential for the spread of tumor cells [18]; these key properties might also enable distant organ involvement in endometriosis. CTCs appear in all major carcinoma groups [17,18] and the enumeration of CTCs in the blood of cancer patients during therapy is considered as a real-time “liquid biopsy” [19] that can serve as a prognostic marker for tumor diseases and therapeutic responsiveness [20]. It is well documented that the expression of EpCAM is frequently down-regulated after epithelial-to-mesenchymal transition (EMT) [21,22,23] and the functional transition of polarized epithelial cells into mobile mesenchymal cells [22]. The isolation of CTCs by relying only on the cell surface expression of EpCAM on CTCs is therefore sub-optimal. We have previously shown that, unlike EpCAM, mHsp70, as detected using the cmHsp70.1 monoclonal antibody (mAb), remains stably expressed on the cell surface of CTCs after EMT [24], and that higher numbers of CTCs can be isolated from the blood of tumor patients using a cmHsp70.1 mAb-based bead approach compared to an equivalent EpCAM mAb-based bead approach [16].

Based on these findings, here, we asked whether free and microvesicle (exosome)-associated Hsp70 in the blood, as well as circulating endometriotic cells (CECs), can serve as potential diagnostic biomarkers for patients with histologically confirmed endometriosis. To address this question, soluble Hsp70 levels were measured using the Hsp70-exo ELISA [25], which is able to quantify free and microvesicle-associated Hsp70, and CECs were enumerated after isolation with cmHsp70.1 mAb- and EpCAM mAb-based bead approaches from patients with endometriosis and extra-uterine endometriosis. Furthermore, the CECs were characterized by immunofluorescence and RNA analyses with a focus on epithelial, mesenchymal and stem cell markers.

Since the diagnosis of endometriosis is often delayed due to non-specific symptoms and presently, only invasive laparoscopic methods can confirm the endometriosis diagnosis, there is a high clinical need for liquid biopsy-based biomarkers. The occurrence of endometriotic cells in extra-uterine organs stimulated us to determine whether tumor biomarkers such as extracellular Hsp70 and the presence of circulating cells in the plasma of endometriosis patients could be potential biomarkers for the diagnosis of the disease.

## 2. Results

### 2.1. Circulating Free and Vesicle-Associated Hsp70 Levels in Patients with Endometriosis, Pre- and Post-Surgery, and Healthy Controls

Free and lipid microvesicle-associated Hsp70 levels in plasma samples from 12 patients with histologically confirmed endometriosis before the start of any therapy were measured using the Hsp70-exo ELISA [25] (Table 1). The mean Hsp70 levels in patients with histologically confirmed endometriosis (mean: 263 ± 345 ng/mL; median: 158 ng/mL) differed from those of a healthy control cohort (n = 108, 35 ng/mL) (Figure 1a), although with large variations (range: 0–1171 ng/mL). The highest soluble Hsp70 level was detected in a patient with lymph node endometriosis (#10: 1171 ng/mL). A subgroup analysis of soluble Hsp70 levels in patients without (n = 7) and with (n = 5) the occurrence of extra-uterine endometriosis revealed significantly elevated Hsp70 levels in patients with lesions outside the uterine cavity compared to healthy controls, whereas those of endometriosis patients without extra-uterine involvement ranged within the normal levels (Figure 1a). 

In a subgroup of three endometriosis patients, blood samples were collected not only before surgery, but also in the follow-up period (after 1 month for patients #11 and #12 and after 6 months for patient #10) post-surgery. A comparison of the soluble Hsp70 levels showed a large drop from 510 ± 589 ng/mL (median: 313 ng/mL) pre-surgery to 38 ± 45 ng/mL Hsp70 after resection of the endometriotic lesions (Figure 1b).

### 2.2. Circulating Endometriotic Cells (CECs) Can Be Isolated from the Blood of Patients with Confirmed Endometriosis by Bead-Based Approaches Targeting Membrane Hsp70 and EpCAM 

The validated cmHsp70.1 mAb- and EpCAM mAb-based bead approaches for isolating circulating tumor cells (CTCs) from EDTA anti-coagulated blood from patients with different tumor entities [16] was used to isolate circulating endometriotic cells (CECs) from patients with histologically confirmed endometriosis (n = 12) and patients with (n = 5) and without extra-uterine endometriosis (n = 7); as a negative control, the EDTA blood of two healthy donors (n = 2) was used in the testing of the isolation approaches. The age of the endometriosis patients ranged from 25 to 48 years (Table 1). As summarized in Table 1, CECs could be isolated from the EDTA blood of all patients (n = 12) with histologically confirmed endometriosis, whereas no cells were isolated from the blood of the healthy individuals (n = 2). The mean number of cells isolated from 7.5 mL of EDTA blood using cmHsp70.1 mAb- and EpCAM mAb-conjugated beads was 21,442 ± 77,711 (median: 258 cells) and 305 ± 418 (median: 178 cells), respectively. The highest CEC yields were obtained from the blood of endometriosis patients with extra-uterine involvement using cmHsp70.1 mAb-conjugated beads for the isolation (Figure 2). Since high CEC counts isolated via cmHsp70.1 mAb beads and elevated soluble Hsp70 levels in the circulation are associated with the occurrence of endometriotic cells outside the uterine cavity, we speculated that these biomarkers might indicate a more aggressive form of endometriosis. To confirm this hypothesis in future clinical studies, both CECs and Hsp70 levels should be determined in the circulation of a larger patient cohort at presentation and in the follow-up period. 

The finding that the elevated soluble Hsp70 levels dropped in all patients after the resection of the endometriotic lesions provides evidence that the Hsp70 in the circulation originates from endometriotic tissues. In line with these findings, the gestational choriocarcinoma cell line JEG-3, which shares some characteristics with primary endometriotic cells, showed strong membrane Hsp70 positivity.

A comparison of the number of CECs in endometriosis patients pre- and 1 month post-surgery indicated a drop in CECs from 890 to 219 (patient #11) and 467 to 123 (patient #12) using the cmHsp70.1 mAb-based bead approach, and from 435 to 384 (patient #11) and 384 to 156 (patient #12) using the EpCAM mAb-based bead approach. Unexpectedly, 6 months post-surgery, the number of CECs isolated using the cmHsp70.1 mAb-based bead approach increased from 12,935 to 186,648 in patient #10 with endometriotic lymph node involvement, but dropped from 379 to 6 when the EpCAM mAb-based bead isolation was used.

### 2.3. Morphological Characterization of Cultured CECs from Patients with Endometriosis Without and with Extra-Uterine Involvement

Freshly isolated CECs from all patients using cmHsp70.1 and EpCAM mAb-conjugated beads were cultured in DMEM medium supplemented with estrogen at 37 °C in 48-well plates. Representative examples of cultured CECs derived from the blood of patient #04 with histologically confirmed endometriosis after 1 week in cell culture, and patients #07 and #10 (both 1 month in cell culture) with endometriosis outside the uterine cavity, which were isolated via mHsp70 targeting (Figure 3a) and EpCAM (Figure 3b) targeting, are illustrated. Interestingly, after 1 week (patient #04) and 1 month (patients #07 and #10) in cell culture, all CECs isolated with EpCAM mAb-based beads lost their viability (Figure 3b), whereas those isolated with cmHsp70.1 mAb beads remained viable and became plastic-adherent (Figure 3a). In the case of blood from a healthy donor, no CECs, but erythrocytes (cmHsp70.1 mAb beads), and an isolation bead (EpCAM mAb beads) were detected after 1 week in cell culture. 

A comparison of the morphology of cmHsp70.1 mAb-isolated cells derived from three endometriosis patients revealed a larger cell diameter (10–20 µm versus 3–5 µm) in the cells of patients #07 and #10 with endometriosis outside the uterine cavity versus patient #04 without extra-uterine involvement. Moreover, the viability in cell culture was much longer (up to 9 months) for mHsp70-expressing CECs derived from patients #7 and #10 with extra-uterine involvement compared to those derived from patient #4 (less than 1 month) without distant organ involvement.

### 2.4. Immunofluorescent Characterization of Isolated CECs from a Patient with Bladder Endometriosis 

The CECs isolated from the blood of patient #07 with endometriotic lesions in the bladder using the cmHsp70.1 mAb-based bead approach were further characterized by immunofluorescence staining. Figure 4a shows the staining of the CECs of patient #07 with Hoechst 33342 and an isotype-matched control antibody labeled with the respective fluorescence dyes (FITC, PE and APC). As illustrated in Figure 4b, cells isolated using cmHsp70.1 mAb-based beads were positively stained for Hoechst 33342 and the typical epithelial marker cytokeratin (green), but were negative for the leukocyte marker CD45 (red). 

To further characterize the biological nature of the circulating cells derived from patient #07, mesenchymal and stemness (CD105, CD44) markers were analyzed using immunofluorescence. As shown in Figure 4b, the cmHsp70.1 mAb-isolated cells were positively stained for Hoechst 33342 and the mesenchymal stemness markers CD105 (purple) and CD44 (green). Due to the limited number of cells, it was only possible to perform immunofluorescence staining of CECs from patient #07.

The presence of the epithelial marker cytokeratin was confirmed by immunofluorescence staining of the JEG-3 gestational choriocarcinoma cell line which shares some characteristics with primary endometriotic cells (Figure 5a). This cell line also expresses mHsp70, the mesenchymal stemness marker CD105, and the epithelial marker CD9, a tetraspanin, but not CD44 on the cell surface of viable cells (Figure 5b,c) (determined by flow cytometry).

### 2.5. RNA Analysis of Isolated CECs from a Patient with Lymph Node Endometriosis 

A high-throughput RNA sequencing and data analysis using the DESeq2 package in R was performed on CECs isolated from patient #10 with lymph node endometriosis using the cmHsp70.1 mAb- and EpCAM mAb-based bead approaches to identify differentially expressed genes. The CECs of patient #10 were chosen because only in the sample from this patient was the number of isolated cells sufficient for these analyses. A total number of 430 genes were found to be differentially regulated (Appendix A, Figure 6) between the EpCAM and cmHsp70.1 mAb-isolated CECs. Table 2 summarizes the results for the RNA analysis of genes in CECs isolated using cmHsp70 mAb- and EpCAM mAb-conjugated beads with an adjusted *p* value (padj) < 0.001 and an absolute log2 fold change value greater than 1.

The differentially expressed genes between CECs isolated with the EpCAM mAb- versus cmHsp70.1 mAb-conjugated bead approach are related to the proto-oncogene Src and kinases (SRC and FGR) [26], cell cycle regulators (RGCC, CDK5RAP3 and CDKN1A), matrix metalloproteinases (MMP2-ASI) [27] (Table 2), which regulate tumor progression and invasiveness via cytoskeletal reorganization and extracellular matrix remodeling, and cell adhesion and migration (ITGAM and ITGAE) in human melanoma [28]. 

In contrast, differentially expressed genes between CECs isolated using the cmHsp70.1 mAb- versus EpCAM mAb-based approach, such as ROCK2 [29], STAT3 [30] and CD82, are related to EMT, stemness and endometrial cell proliferation [31] (Table 2), which are regulated via up-regulated CD44/STAT3 signaling in tumor cells [32]. Down-regulated STAT3 signaling has been associated with fewer endometriotic lesions in an endometriotic mouse model [33] and in human endometriotic tissues [34]. A potential link between STAT3 and EMT/stemness markers is suggested by that observation that the mHsp70 CECs were positive for the markers CD44 and CD105 (Figure 4b). 

## 3. Discussion

As invasive methods are currently the only option for diagnosing endometriosis [1,2,3], there is a high unmet clinical need for the establishment of reliable liquid biopsy-based biomarkers that can improve the diagnosis of the disease. Since endometriosis is a disease with occurrence in extra-uterine organs such as the brain, lung, bladder, rectum or lymph nodes [1,2,3], and endometriosis-affected tissues share some pathological characteristics with malignant tumors (migration, invasiveness, inflammation, cell growth, angiogenesis, cell adhesion, cytokine production [35]), this pilot study examined the potential of soluble, microvesicle- and membrane-associated Hsp70 as diagnostic targets. We showed that the levels of free and microvesicle-associated Hsp70 in the circulation of patients with confirmed endometriosis were higher than those of a healthy control cohort, although with large variations. With a mean soluble Hsp70 concentration of 375.8 ng/mL, the levels in endometriosis patients with extra-uterine involvement were comparable to those of patients with lung, brain and breast carcinomas [12,13,14,15,16,36] and were significantly higher than those of the healthy control group (35 ng/mL); meanwhile, those of patients without extra-uterine involvement ranged within the normal levels. This finding supports the hypothesis that soluble Hsp70 might play a role in endometriosis that spreads outside the uterine cavity and might indicate a more aggressive form of the disease. To confirm this hypothesis, studies with larger patient cohorts and longer follow-up periods are urgently needed.

Previous studies have indicated that elevated Hsp70 levels in the blood are associated with an unfavorable tumor prognosis and a higher risk for developing distant metastases in different mammalian species [15,36,37]. In our small cohort, we demonstrated that the highest soluble Hsp70 levels were found in patients with lymph node, bladder and rectum endometriosis, indicating that an involvement of extra-uterine organs might be associated with a more severe disease. Protein expression profiling of ovarian cancer and the development of a prognostic model based on univariate and multiple Cox regression analyses has shown that Hsp70 expression can predict the survival prognosis of patients with ovarian cancer. The risk model exhibited a higher predictive power than age, tumor grade and tumor stage [38]. Hsp70 expression has also been shown to predict the prognosis and chemotherapy response of patients with epithelial ovarian cancer [39]. Further insights into the involvement of Hsp70 in the progression of ovarian cancer have been provided by studies demonstrating that the inhibition of Hsp70 nuclear translocation and interruption of its interaction with Notch1’s intracellular domain reduces the migratory and invasive capacity of the OVCAR3 serous ovarian cancer cell line and the growth of OVCAR3-derived tumors in vivo [40]. The Notch pathway, which is involved in proliferation, differentiation and cell survival, is one of the most activated pathways in many cancers, including ovarian [41] and endometrial [42] cancers. Furthermore, stress-induced phosphoprotein-1 (STIP-1), an adaptor protein for Hsp70 that is up-regulated in the circulation of patients with endometriosis, is known to regulate cell migration via matrix metalloproteinase 9 (MMP-9) [43] and metastasis [44]. Elevated levels of the co-chaperone Hsp70 might therefore be involved in the dissemination of endometrial cells to distant organs and in the development of endometriosis-related ovarian and endometrial cancers. However, further studies in a larger number of patients are required in order to better understand the relationship(s) between the levels of soluble Hsp70 and clinical status.

It has been demonstrated that only tumor cells expressing mHsp70 actively release Hsp70 into extracellular lipid vesicles with proven biophysical characteristics of exosomes [11], and that high Hsp70 levels in the blood are associated with therapy resistance in patients with cancer [25,45]. Moreover, the density of mHsp70 expression on cancer metastases appears higher than that on the corresponding primary tumor [26]. Since the levels of circulating Hsp70 are comparable in patients with malignant tumors and endometriosis that occurs in extra-uterine organs including the lymph nodes, rectum and bladder, we speculate that the soluble Hsp70 might originate from the cell membrane of endometriotic cells. This hypothesis is supported by the finding that soluble Hsp70 levels dropped drastically after laparoscopic resection of endometriotic lesions, similar to the reduced levels in tumor patients after successful therapy that decreased the tumor size [46]. Due to the lack of freshly resected tissue and the limited availability of CECs, the final proof for mHsp70 positivity on endometriotic cells could not be provided in this study. However, strong mHsp70 positivity was demonstrated for the gestational choriocarcinoma cell line JEG-3, which has certain characteristics that resemble those of endometriotic cells, but is nevertheless a tumor cell. Future studies with larger patient cohorts and access to resected endometriotic tissue are necessary to confirm mHsp70 expression on endometriotic cells.

Since the exact mechanism through which endometriotic cells invade into extra-uterine organs has not yet been elucidated, we asked whether CECs might be involved in this process. Therefore, we isolated CECs from EDTA anti-coagulated blood of patients with confirmed endometriosis and extra-uterine involvement using a bead-based approach, which was applied for the isolation of CTCs by targeting mHsp70 or EpCAM through employing cmHsp70.1 mAb- and EpCAM mAb-conjugated beads, respectively. Interestingly, CECs could be isolated with both mAb-conjugated bead systems from all blood samples of the endometriosis patients, with drastically higher cell yields using mHsp70 as the selection target and in patients with extra-uterine endometriosis. These data indicate that the origins of CECs isolated via membrane-bound Hsp70 versus EpCAM differ. Moreover, CECs isolated from patients with extra-uterine endometriosis using cmHsp70.1 mAb beads, in contrast to EpCAM mAb beads, could be cultured, remaining viable for several months, and were positive for the typical CTC marker cytokeratin and the mesenchymal stemness markers CD44 and CD105, but were negative for the leukocyte marker CD45. These CTC characteristics might explain the occurrence of endometriotic cells in distant organs, since these cells are essential in the metastatic spread in tumor patients [17,18,47].

The morphology of the CECs from the patients with endometriotic bladder and lymph nodes isolated using cmHsp70.1 mAb beads differed from those of the other endometriosis patients as they were rounder in shape and larger in size. Besides the altered morphology, these CECs tended to grow in cell clusters and could be propagated in cell culture for a longer period of time (up to 9 months). Due to these differences and due to their molecular features, we propose that the cells isolated using cmHsp70.1 mAb beads from the patient with endometriotic bladder and lymph nodes could be considered to be “CTC-like”. However, it has yet to be demonstrated whether these characteristics are related to or explain the perceived relationship(s) between endometriosis and gynecological cancers [48,49].

It is known that EpCAM expression on the surface of CTCs is reduced after EMT, whereas mHsp70 remains stably expressed after EMT [19,20,21,22,24]. Since the cell yield after isolation with cmHsp70.1 mAb-conjugated beads was much higher than after isolation with EpCAM mAb-conjugated beads, we hypothesize a mesenchymal origin for the circulating CECs with mHsp70 as a target structure. To further examine the possible mesenchymal characteristics of cells isolated using the cmHsp70.1 mAb-conjugated beads from patients with confirmed endometriosis, isolated CECs were immunofluorescently stained for typical mesenchymal and stem cell markers such as CD105 and CD44 as well as for the epithelial marker cytokeratin. The gestational choriocarcinoma cell line JEG-3 served as a control for the validation of the CTC markers. Positive staining of the isolated CECs for the mesenchymal stemness markers CD44, a hyaluronic acid receptor, and CD105, a co-receptor for TGF-β, indicate that these cells have stem cell characteristics [50]. Together with positive staining for cytokeratin [51] and mHsp70, the cultured endometriotic cells share some characteristics with tumor cells, which supports the hypothesis that the CECs isolated from endometriosis patients have malignant traits. This is in line with the finding that CECs isolated based on mHsp70 expression remained viable in cell culture for long periods of time, whereas those isolated based on their expression of EpCAM lost viability in cell culture.

The CECs isolated from patient #10 with endometriotic lymph nodes based on EpCAM and mHsp70 expression were profiled to identify differentially regulated genes. The differentially expressed genes of the CECs isolated with EpCAM mAb-conjugated beads were found to be related to proto-oncogenes, cell cycle regulation [26], cytoskeletal reorganization, extracellular matrix remodeling [27] and cell adhesion [35]. In the cmHsp70.1 mAb-isolated CECs, the differentially expressed genes were related to markers for EMT [52,53], stemness [54] and endometrial cell proliferation, which are regulated via an up-regulation of CD44 and STAT3 signaling [31]. These data are in line with the positive immunofluorescence staining of mHsp70-expressing CECs for CD44 (Figure 4b). Although not significantly different, the expression of genes encoding for NCAM1 (CD56), a neuronal cellular adhesion molecule that is described as a signal transducer for the regulation of cell growth, migration, proliferation, apoptosis and differentiation [55] and reflects pathological features of tumor cells [35], as well as additional EMT markers such as ALCAM, ITGB1 and ITGA6 [55,56,57] were also differentially expressed in cmHsp70.1 bead-isolated CECs. These findings support the assumption of pathological characteristics in circulating cells that were isolated using cmHsp70.1 mAb-conjugated beads.

## 4. Materials and Methods

### 4.1. Sample Collection

This pilot study analyzed EDTA anti-coagulated blood (2 × 7.5 mL) from 12 therapy-naïve, Caucasian female patients with an age range of 18–50 with histologically confirmed endometriosis without and with extra-uterine involvement who were treated at the Department of Gynecology and Obstetrics, TUM University Hospital, School of Medicine and Health, Technical University Munich (TUM), between September 2022 and January 2024. Post-menopausal patients and patients with known malignancies of any kind were excluded from the study. Age-matched healthy donors were included as controls. Blood was collected from the patients with endometriosis prior to surgery. In 3 patients, additional blood samples were collected before and 1 and 6 months after resection of endometriotic lesions by laparoscopy. The study was approved by the Ethical Committee of the TUM University Hospital, School of Medicine and Health (2023-407-S-NP). Before starting the study, written informed consent was obtained from all the patients and normal donors. The study was performed in accordance with the guidelines of the 1975 Declaration of Helsinki.

### 4.2. Cell Culture

The human cancer cell line JEG-3 (RRID:CVCL_0363; gestational choriocarcinoma), kindly provided by the group of Prof. Andreas Pichlmair (Institute of Virology, TUM), was cultured in RPMI Medium (Gibco Thermo Fisher Scientific, Waltham, MA, USA), supplemented with 10% *v*/*v* heat-inactivated fetal bovine serum (FBS; Sigma-Aldrich, St. Louis, MO, USA) and antibiotics (100 IU/mL penicillin and 100 mg/mL streptomycin; Sigma-Aldrich, St. Louis, MO, USA), at 37 °C with 5% *v*/*v* CO_2_ in a humidified atmosphere. The cell line was shown to be positive for mHsp70 expression by flow cytometry and immunofluorescence staining.

### 4.3. Measurement of Circulating Free and Lipid Vesicle-Associated Hsp70 Levels Using the Hsp70-Exo ELISA

For plasma preparation, EDTA anti-coagulated whole blood (S-Monovette, Sarstedt, Nürmbrecht, Germany) was centrifuged at 1500× *g* for 15 min at room temperature and aliquots of plasma (300 µL) were stored at −80 °C. For the Hsp70-exo ELISA, 96-well MaxiSorp Nunc-Immuno plates (Thermo, Rochester, NY, USA) were coated with the cmHsp70.2 coating mAb (1 µg/mL; multimmune GmbH, Munich, Germany) in sodium carbonate buffer (0.1 M sodium carbonate, 0.1 M sodium hydrogen carbonate, pH 9.6; Sigma-Aldrich (Darmstadt, Germany)) overnight at room temperature. After rinsing the plates with washing buffer (phosphate-buffered saline (PBS), Life Technologies, Darmstadt, Germany) containing 0.05% *v*/*v* Tween-20 (Calbiochem, Merck, Darmstadt, Germany), the plates were blocked with liquid plate sealer (Condor Bioscience GmbH, Wangen i. Allgäu, Germany) for 30 min to prevent non-specific binding. The plates were washed and plasma samples (100 µL) and an 8-point Hsp70 standard (0–100 ng/mL), both diluted 1:5 in StabilZyme Select Stabilizer (Diarect GmbH, Freiburg i. Breisgau, Germany), were added. After 30 min of incubation and washing, the biotinylated cmHsp70.1 detection mAb (multimmune GmbH, Munich, Germany; 200 ng/mL) dissolved in HRP-Protector^TM^ (Candor Biosciences GmbH, Wangen i. Allgäu, Germany) was added for 30 min, after which, the plates were rinsed with washing buffer. After another incubation for 30 min with 57 ng/mL Streptavidin (Senova GmbH, Weimar, Germany) in HRP-Protector^TM^ (Candor Biosciences GmbH, Wangen i. Allgäu, Germany), the plates were incubated with the substrate reagent (100 µL) (BioFX TMB Super Sensitive One Component HRP Microwell Substrate, Surmodics, Inc., Eden Prairie, MN, USA) for 15 min, after which, the reaction was stopped with 2 N H_2_SO_4_ (50 µL) and the absorbances were measured at 450 nm using a microplate reader (VICTOR X4 Multilabel Plate Reader, PerkinElmer, Waltham, MA, USA). The absorbance readings were corrected with the absorbance at 570 nm [25]. The specificity of the Hsp70-exo ELISA antibodies cmHsp70.1 and cmHsp70.2, which detect free and lipid microvesicle-bound Hsp70, was confirmed by Western blot and dot blot analyses using different stress proteins and different members of the HSP70 family, and by experiments that spiked lipid-bound Hsp70 into the blood of healthy human individuals [25]. 

### 4.4. Isolation of Circulating Cells with cmHsp70.1 and EpCAM Antibody-Coupled S-pluriBeads

The isolation of circulating CECs from patients with endometriosis and healthy donors was performed according to a method described previously for CTCs [16,24]. Briefly, EDTA anti-coagulated blood (7.5 mL) was incubated with S-pluriBeads (PluriSelect Life Sciences, Leipzig, Germany), which were covalently coupled to the cmHsp70.1 (multimmune GmbH, Munich, Germany) or EpCAM (CD326, clone HEA 125; Origene/Acris GmbH, Herford, Germany) mAbs for 30 min under gentle rotation at room temperature. To reduce the number of erythrocytes in the sample, cells bound to the antibody-coupled beads were washed on a 30 µm pluriStrainer^®^ with washing buffer (PluriSelect Life Sciences, Leipzig, Germany). CECs were detached from the beads by incubating them with detachment buffer (PluriSelect Life Sciences, Leipzig, Germany) for 10 min at room temperature. The sample was then filtered through a sterile strainer, washed twice with medium, and incubated overnight at 37 °C in a 48-well plate. After 24 h and again after 1 week, the cells were counted using a Zeiss Axiovert microscope (Zeiss, Oberkoch, Germany). The isolated cells were kept in culture (DMEM Medium; Gibco Thermo Fisher Scientific, Waltham MA USA) supplemented with 1 nM β-estrogen (Invitrogen Merck, Darmstadt, Germany), 10% *v*/*v* heat-inactivated FBS (Sigma-Aldrich, St. Louis, MO, USA) and antibiotics (100 IU/mL penicillin and 100 mg/mL streptomycin; Sigma-Aldrich, St. Louis, MO, USA) for further analysis.

### 4.5. Immunohistochemical Staining 

Adherent cells were cultivated in an 8-well chamber slide (Invitrogen, Thermo Fisher Scientific, Waltham MA, USA). After three washing steps in flow cytometry buffer (PBS; Life Technologies) containing 10% *v*/*v* FBS (Sigma Aldrich, St. Louis, MO, USA), the cells were incubated with the following antibodies at room temperature for 30 min in the dark: a cytokeratin cocktail, CK7 (LP5K, Millipore, Billerica, MA, USA), CK19 (A53-B/A2, Exbio, Vestec, Czech Republic), panCK (C11, Exbio, Vestec, Czech Republic); CD45-Alexa Fluor^®^ 488 (A4-120-C100, Exbio, Vestec, Czech Republic); CD105-PE (42A3, BioLegend, San Diego, CA, USA); and CD44-FITC (C44Mab-5, BioLegend, San Diego, CA, USA). All the antibodies were diluted in PBS (1:50). After another wash, the nuclei were counter-stained with Hoechst 33342 (1:1000; Invitrogen Thermo Fisher Scientific, Waltham, MA, USA) for 5 min at room temperature. The cells were imaged using a Zeiss Axio Observer.Z1 microscope (Zeiss, Jena, Germany) at the magnifications indicated and image analysis was performed using AxioVision SE64 Rel.4.9. Multi-color images were produced by merging. 

### 4.6. Flow Cytometry 

JEG-3 cells were trypsinized and the single-cell suspensions were washed twice with flow cytometry buffer (PBS; Life Technologies, containing 10% *v*/*v* FBS, Sigma Aldrich). The cells (100,000 each) were then incubated with the following fluorescently labeled antibodies in the dark for 30 min on ice: cmHsp70.1-FITC (multimmune GmbH, Munich, Germany); CD44-FITC (C44Mab-5, BioLegend, San Diego, CA, USA); CD9-PE (H19a, BioLegend, San Diego, CA, USA); CD105-PE (43A3, BioLegend, San Diego, CA, USA). After a final washing step, the cells were resuspended in flow cytometry buffer and analyzed using a BD FACSCalibur™ flow cytometer (BD Biosciences, Franklin Lakes, NJ, USA). Immediately before the flow cytometric analysis, propidium iodide (1 µL/100 µL; Sigma-Aldrich, St. Louis, MO, USA) was added to determine cell viability. Only PI-negative (viable) cells were gated and analyzed. An isotype-matched control antibody (BD Biosciences, Franklin Lakes, NJ, USA) was used to determine non-specific staining. 

### 4.7. RNA Preparation and Analysis 

Library preparation for the bulk sequencing of poly(A)-RNA was performed as described previously [58]. Barcoded cDNA of each sample was generated using the Maxima RT polymerase (Thermo Fisher Scientific, Los Angeles, CA, USA) and oligo-dT primers containing barcodes, unique molecular identifiers (UMIs) and an adaptor. The 5′-ends of the cDNAs were extended by a template switch oligo (TSO) and the full-length cDNA was amplified with primers that bind to the TSO site and the adaptor. An NEBNext^®^ Ultra^®^ kit was used to fragment the cDNA. After end repair and A-tailing, a TruSeq adaptor was ligated and the 3′-end fragments were finally amplified using primers with Illumina P5 and P7 overhangs. The library was sequenced using a NextSeq 500 (Illumina, Los Angeles, CA, USA) with 65 cycles for the cDNA in read 1 and 19 cycles for the barcodes and UMIs in read 2. The data were processed using the published Drop-seq pipeline (v1.0) to generate sample- and gene-wise UMI tables [59]. The reference genome GRCh38 was used for alignment. The transcript and gene definitions of GENCODE v38 were used. 

The raw sequencing reads were subjected to quality checks using FastQC (version 0.11.9) and were processed with Trimmomatic to trim low-quality bases and to remove adaptor sequences in FASTQ format. These clean paired-end reads were aligned to the human reference genome (GRCh38) using HISAT2 (version 2.0.5) with the default parameters. Gene expression levels were quantified using featureCounts, generating raw count data that were subsequently normalized using DESeq2 which accounts for variations in sequencing depth and other technical factors. The DESeq2 R package in the Bioconductor platform was employed to identify differentially expressed genes (DEGs). DESeq2 filters genes based on their average expression across all samples, excluding those with mean normalized counts below a threshold that is automatically determined to maximize the number of detected genes, depending on user-defined criteria. Statistically significant DEGs were identified by comparing the expression levels of cmHsp70.1 mAb- and EpCAM mAb-isolated CEC samples, with the significance criteria set at an adjusted *p* value (padj) < 0.001 and an absolute log2 fold change greater than 1 (>1) [60]. 

To compare and analyze the RNA-seq results, we utilized multiple comprehensive and publicly available genomic and proteomic datasets. These resources provided a broad spectrum of gene expression profiles across various cancer types, enabling robust validation and cross-referencing of our findings. Among the identified DEGs that were notably upregulated in mHsp70- and EpCAM-positive CECs were genes which are associated with epithelial-to-mesenchymal transition (EMT) pathways, stemness, proliferation, matrix remodeling, cell cycle regulation and adhesion.

### 4.8. Statistical Evaluations

For the statistical analysis of all dependent samples, the Wilcoxon test (GraphPad Prism 10) was used, and for all independent samples, the Kruskal–Wallis test was performed, considering a 5% significance level.

## 5. Conclusions

The occurrence of endometriotic cells in extra-uterine organs stimulated us to determine whether extracellular Hsp70 concentrations and the presence and prevalence of CECs in the plasma of endometriosis patients could be potential biomarkers for a minimally invasive diagnosis of the disease. Similar to observations from tumor patients, we demonstrated significantly elevated soluble Hsp70 levels and the presence of CECs with characteristics of CTCs in the blood of patients with endometriosis outside the uterine cavity. These data might be indicative of a more aggressive form of endometriosis with extra-uterine involvement. The data warrant further studies to determine the soluble Hsp70 levels and CEC counts in larger patient cohorts at presentation and in the follow-up period and with access to resected endometriotic tissues.

## Figures and Tables

**Figure 1 ijms-25-11643-f001:**
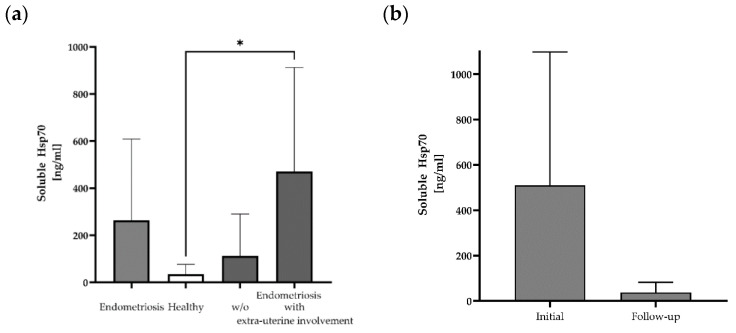
Soluble Hsp70 levels in the blood of patients with endometriosis and healthy controls. Free and lipid vesicle-associated Hsp70 levels [ng/mL] in the plasma of patients with histologically confirmed endometriosis (n = 12), a healthy cohort (n = 108) and a subgroup of endometriosis patients without (w/o) (n = 7) and with extra-uterine involvement (n = 5) (* *p* < 0.05 (Kruskal–Wallis test)) (**a**), and selected patients with endometriosis pre- (initial) and post-surgery (follow-up) (n = 3) (**b**), as determined using the Hsp70-exo ELISA.

**Figure 2 ijms-25-11643-f002:**
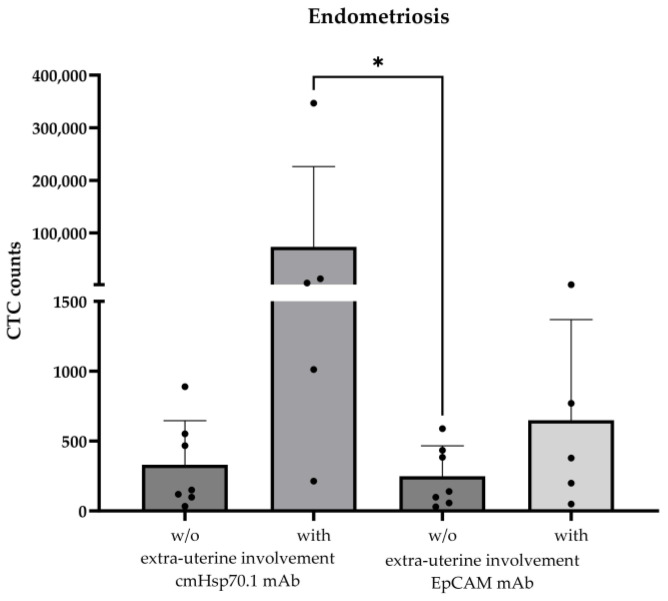
Comparison of cmHsp70.1 mAb- and EpCAM mAb-isolated circulating endometriotic cells (CECs) in the blood of patients without (w/o) (n = 7) and with (n = 5) endometriotic lesions in extra-uterine organs. CECs were derived from EDTA anti-coagulated blood (7.5 mL) of patients with histologically confirmed endometriosis (n = 12); * *p* < 0.05 (Kruskal–Wallis test).

**Figure 3 ijms-25-11643-f003:**
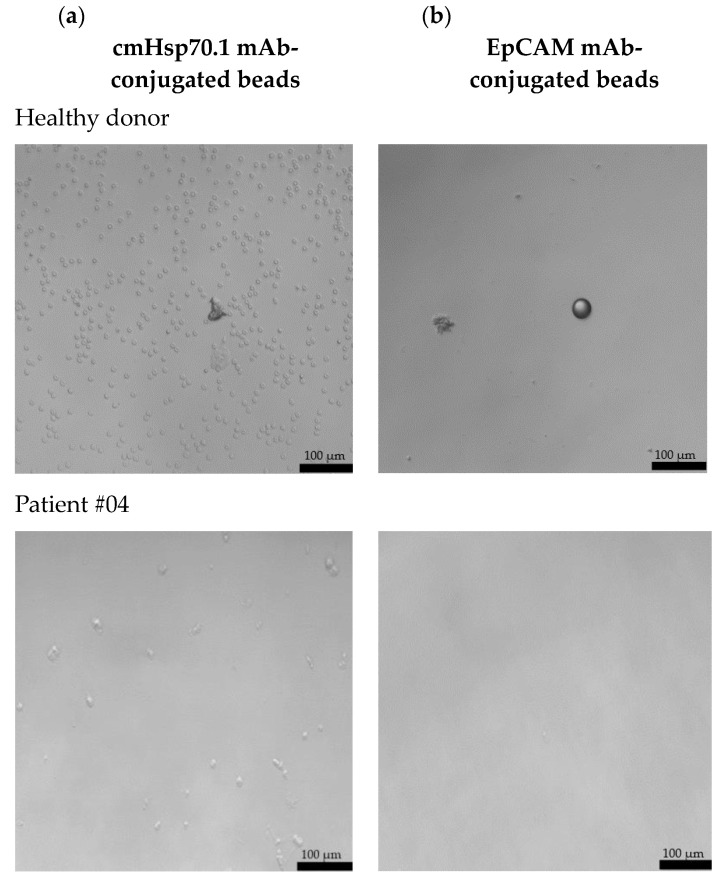
Morphology of cultured CECs isolated from the EDTA anti-coagulated blood (7.5 mL) of a healthy donor, patient #04 with histologically confirmed endometriosis (1 week in cell culture) and patients #07 and #10 with confirmed endometriosis and distant organ involvement (1 month in cell culture) using cmHsp70.1 (**a**) mAb- and EpCAM (**b**) mAb-conjugated beads. Magnification 20×; scale bar: 100 µm.

**Figure 4 ijms-25-11643-f004:**
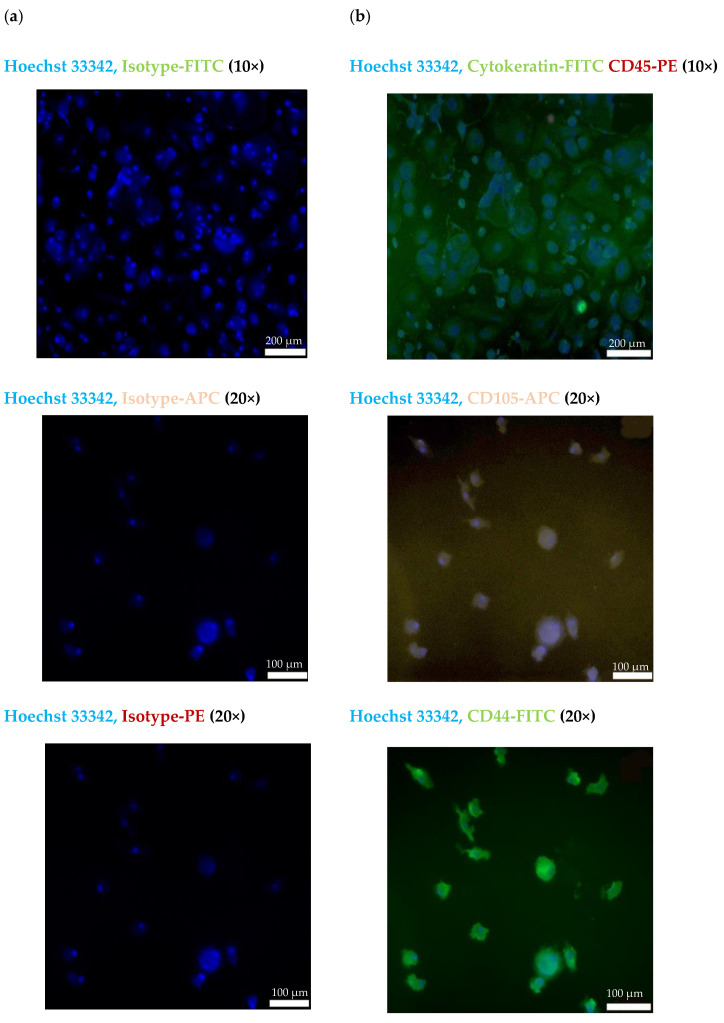
Representative immunofluorescence photomicrographs of cultured CECs using a Zeiss Axio Observer.Z1 microscope. Immunofluorescent images of isolated cells from endometriosis patient #07 using cmHsp70 mAb-conjugated beads after culturing for 7 days. (**a**) CECs stained with Hoechst 33342 (1:1000) and an isotype-matched control antibody conjugated either with FITC, PE or APC. (**b**) CECs stained with Hoechst 33342 (1:1000) and a FITC-labeled cytokeratin cocktail (green) and PE-labeled CD45 (red); Hoechst 33342 (1:1000) and CD105-APC (purple); and Hoechst 33342 (1:1000) and CD44-FITC (green). All antibodies were diluted 1:50. Magnification: 10×, 20×; scale bars: 200 µm, 100 µm.

**Figure 5 ijms-25-11643-f005:**
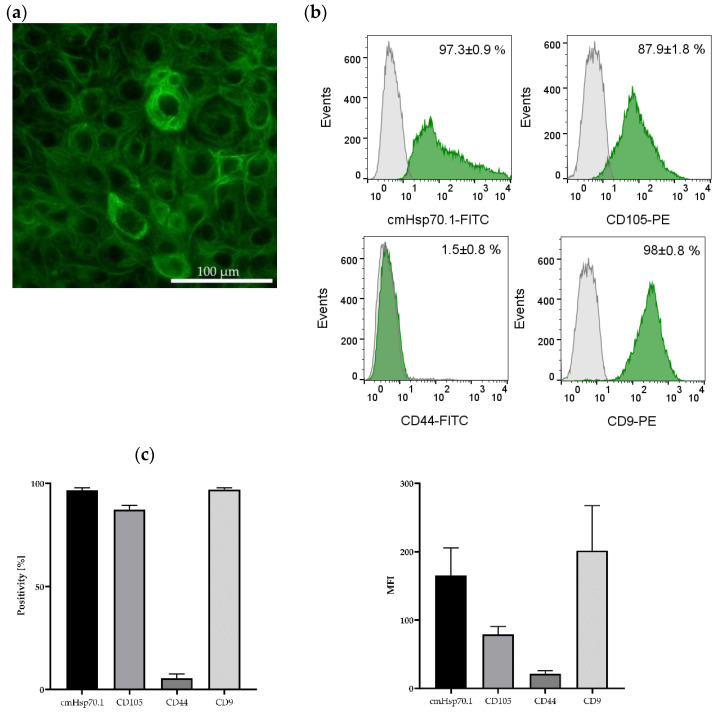
Immunofluorescence and flow cytometric analysis of JEG-3 cells. The gestational choriocarcinoma cell line JEG-3 was stained with Hoechst 33342 (1:1000) and a FITC-labeled cytokeratin cocktail (1:50). Magnification: 20×; scale bar: 50 µm (**a**). Representative histograms of mHsp70, CD105, CD44 and CD9 expression on viable JEG-3 cells after staining with cmHsp70.1-FITC, CD105-PE, CD44-FITC and CD9-PE mAbs; the white histograms represent the isotype-matched control, the green histograms represent the specific stainings with the indicated mAbs (**b**). Percentages of positively stained JEG-3 cells (Positivity, [%]) and mean fluorescence intensity (MFI) values from 3 independent experiments ± standard deviation (**c**).

**Figure 6 ijms-25-11643-f006:**
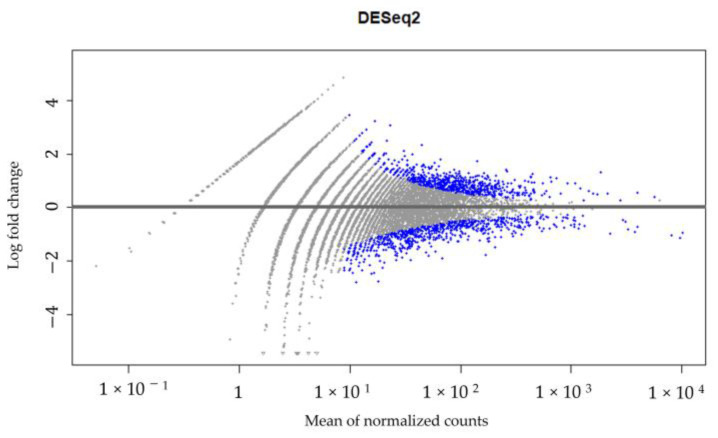
DESeq2 results. Gene expression in CECs isolated using cmHsp70.1 mAb- and EpCAM mAb-conjugated beads from patient #10 with histologically confirmed lymph node endometriosis.

**Table 1 ijms-25-11643-t001:** Enumeration of circulating endometriotic cells (CECs). Numbers of isolated CECs using the cmHsp70.1 and EpCAM mAb-based isolation techniques on EDTA anti-coagulated blood (7.5 mL) of patients with histologically confirmed endometriosis (n = 12) and two healthy donors (HD). Patients #02, #06, #07 and #08 had involvement of the bladder and #10 had endometriotic lymph nodes. Additional EDTA blood samples were obtained from patients #11 and #12 before and 1 month post-surgery, and from patient #10 before and 6 months post-surgery.

Endometriosis	Age	Distant Organ Involvement	Bead Functionalization	
Patient ID			cmHsp70.1 mAb	EpCAM mAb
01	25	No	150	589
02	25	Bladder	1013	1845
03	48	No	120	98
04	34	No	552	30
05	45	No	34	57
06	44	Bladder, rectum	4963	770
07	27	Bladder	346,896	50
08	36	Bladder	214	199
09	33	No	98	139
10	42	Lymph node	12,935	379
10: follow-up			186,648	6
11	47	No	890	435
11: follow up			219	384
12	29	No	467	384
12: follow up			123	156
Healthy donors				
01	25	No	0	0
02	25	No	0	0

**Table 2 ijms-25-11643-t002:** Differentially expressed genes between CECs isolated using EpCAM mAb- and cmHsp70.1 mAb-conjugated beads from patient #10 with histologically confirmed endometriotic cell-infiltrated lymph nodes. Expression levels of statistically significant differentially expressed genes (DEGs) between cmHsp70.1 mAb- and EpCAM mAb-isolated CECs with an adjusted *p* value (padj) < 0.001 and an absolute log2 fold change greater than 1.

Gene	Function	Base Mean	Log2 Fold Change	Padj
EpCAM mAb				
*SRC*	Progression Cytoskeletal reorganization	44	1.1	4 × 10^−3^
*FGR*	Src kinase	230.5	1.1	2 × 10^−13^
*MMP2-AS1*	Extracellular matrix remodeling	11.8	1.7	4 × 10^−3^
*CDK5RAP3*	Cell cycle regulation	10.2	2.1	7 × 10^−4^
*CDKN1A*	Cell cycle regulation	51.9	1.3	1 × 10^−4^
*RGCC*	Cell cycle regulation	94.8	1.3	3 × 10^−8^
*ITGAM*	Cell adhesion and migration	20	1	1 × 10^−4^
*ITGAE*	Cell adhesion and migration	29.5	1.4	1 × 10^−3^
cmHsp70.1 mAb				
*ROCK2*	EMT: Cytoskeletal organization	66.2	−1.8	4 × 10^−5^
*STAT3*	EMT: Stemness-related marker	107.8	−1	1 × 10^−3^
*CD82*	Cell adhesion	134.2	−1	2 × 10^−4^

## Data Availability

The original contributions presented in the study are included in the manuscript and in the Appendix A; further inquiries can be directed to the corresponding author.

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
