# Peer review of "Extracellular Hsp70 and Circulating Endometriotic Cells as Novel Biomarkers for Endometriosis"

_ijms, 2024, doi:10.3390/ijms252111643_

Round 1
Reviewer 1 Report (New Reviewer)
Comments and Suggestions for Authors
See the attachment.

Author Response
Reviewer 1
This research presents meaningful results overall. I would like to provide the
following comments to improve the quality of the paper.
(Comment 1) There are several instances of outdated references. It is necessary to
update the references to papers published after 2010 overall. Additionally, please
do not include references in the sub-title, and keep the sub-title concise.
Introduction Section
(Response 1)
16 References (older than 2010) were deleted as recommended and references were deleted from the sub-titles.
(Comment 2) The Introduction Section should be based on references. Please
include the references at the end of any sentences where they are missing.
- e.g. Worldwide, 10-15% of all women are affected by endometriosis, an
estrogen-dependent benign gynecological disease [reference].
(Response 2)
References have been included to the introduction Section as recommended.
(Comment 3) Please conclude the last sentence of the introduction with the
research objectives and hypotheses.
(Response 3)
Research objectives and hypothesis have been included as a last sentence in the introduction as follows.
“Since the diagnosis of endometriosis is often delayed due to non-specific symptoms and presently only invasive laparoscopic methods can confirm the diagnose endometriosis there is a high clinical need for liquid biopsy-based biomarkers. The occurrence of endometriotic cells in extra-uterine organs stimulated us to determine tumor biomarkers such as extracellular Hsp70 and the presence of circulating cells in the plasma of endometriosis patients as potential biomarkers for the diagnosis of the disease.”
Result Section
(Comment 4) The Results section is a flaw in this study. The Results section should
present only the facts based on the results. Please revise the section to exclude
unnecessary content, including the sentence below.
- Since endometriosis is generally considered as a non-malignant disease and
soluble Hsp70 predominantly originates from malignant tumor cells expressing
Hsp70 on their plasma membrane, it was expected that Hsp70 levels in the
circulation of endometriosis patients would be within normal levels.
- Unexpectedly,
- A subgroup analysis of soluble Hsp70 levels in patients without (n=7) and with
(n=5) occurrence of extra-uterine endometriosis revealed significantly elevated
Hsp70 levels in patients with lesions outside the uterine cavity compared to
healthy controls that are comparable to those measured in patients with
malignant tumors such as lung, brain and breast carcinomas [18-21,39]. In
contrast, soluble Hsp70 levels in the circulation of endometriosis patients
without an extra-uterine involvement ranged within normal levels (Figure 1a).
Therefore, we hypothesize that increased soluble Hsp70 levels might indicate a
more aggressive form of the disease which spreads outside the uterine cavity.
To confirm this hypothesis, studies with larger patient cohorts and longer
follow-up periods are urgently needed.
- This finding suggests that soluble Hsp70 originates from endometriotic cells expressing mHsp70.
(Response 4)
The sentences mentioned above have been deleted from the results part and included in the discussion part.
The authors want to thank for the helpful comments of reveiwer 1.
Reviewer 2 Report (New Reviewer)
Comments and Suggestions for Authors
The manuscript by Guder et al. entitled ‘Extracellular Hsp70 and circulating endometriotic cells as novel biomarkers for endometriosis” suggests novel markers of endometriosis. The study seems to be interesting. However, its obvious limitation is a small sample size. Hsp70 overexpression is a common stress-associated response in cells. Its elevation is commonly observed in multiple diseases. Do the authors consider determination of its circulating levels specific enough? How the specificity was tested?
- Heat shock protein 70 is abbreviated as either Hsp70 or HSP70. Please select one abbreviation and apply it throughout the manuscript
- ER: provide the full name when mentioned for the first time
- Please provide the version of GraphPad Prism used for statistical analysis
- Provide demographic data on patients. Which exclusion criteria were applied? Multiple conditions can be associated with elevated levels of Hsp70
- Figure 1. Provide the information on statistical tests and the way numerical data are demonstrated, as well as the meaning of a * sign, in the legend
- Limitations are not mentioned.
Author Response
Reviewer 2
(Comment 1)
The manuscript by Guder et al. entitled ‘Extracellular Hsp70 and circulating endometriotic cells as novel biomarkers for endometriosis” suggests novel markers of endometriosis. The study seems to be interesting. However, its obvious limitation is a small sample size. Hsp70 overexpression is a common stress-associated response in cells. Its elevation is commonly observed in multiple diseases. Do the authors consider determination of its circulating levels specific enough? How the specificity was tested?
(Response 1)
The Hsp70-exo ELISA which was used to determine soluble Hsp70 detects exosomal Hsp70 originating from membrane Hsp70 positive cells and free Hsp70 originating from dead cells. The specificity of both antibodies (cmHsp70.1 and cmHsp70.2) of the Hsp70-exo sandwich ELISA has been determined by Western blot analysis using different stress proteins and different members of the HSP70 family and by spiking experiments using liposomal-bound Hsp70 in the blood of healthy individuals. These published data (Werner et al. 2021) are cited in the Materials and Methods part.
As mentioned by the reviewer, elevated Hsp70 levels are occurring in different diseases including inflammation. A comparison of soluble Hsp70 levels in the blood of patients with chronic hepatitis, liver cirrhosis and hepatocellular carcinomas revealed highest levels in tumor patients (Gehrmann et al. 2014). A comparison of the amount of free and lipid microvesicle-bound (exosomal) Hsp70 in the blood of tumor patients using two different ELISA systems (R&D, specific for free Hsp70 and Hsp70-exo ELISA, specific for free and lipid microvesicle-bound Hsp70) revealed significantly higher Hsp70 levels (>40-fold) of lipid microvesicle-bound Hsp70 (Lennartz et al. 2023). Since the levels of soluble Hsp70 are comparable in patients with endometriosis and tumor patients we assume that in endometriosis patients with extra-uterine involvement predominantly exosomal Hsp70 is responsible for the elevated Hsp70 levels.
(Comment 2)
- Heat shock protein 70 is abbreviated as either Hsp70 or HSP70. Please select one abbreviation and apply it throughout the manuscript
(Response 2)
The spelling of the abbreviations HSP70 and Hsp70 is correct. HSP70 in capital letters means the family of all HSP with a molecular weight of approximately 70 kDa and Hsp70 means the stress-inducible 72 kDa stress protein which is a member of the HSP70 family.
(Comment 3)
- ER: provide the full name when mentioned for the first time
(Response 3)
ER has been spelled as “endoplasmic reticulum” when it was mentioned first.
(Comment 4)
- Please provide the version of GraphPad Prism used for statistical analysis
(Response 4)
GraphPad Prism version 10 has been used.
(Comment 5)
- Provide demographic data on patients. Which exclusion criteria were applied? Multiple conditions can be associated with elevated levels of Hsp70
(Response 5)
Demographic data on patients and exclusion criteria have been included in the Materials and Methods part. The conditions which can be associated with elevated levels of Hsp70 have been outlined.
(Comment 6)
- Figure 1. Provide the information on statistical tests and the way numerical data are demonstrated, as well as the meaning of a * sign, in the legend
(Response 6)
The *p value (p<0.05; p=0.020) has been included as well as the statistical test.
(Comment 7)
- Limitations are not mentioned.
(Response 7)
- As recommended, the major limitation of the study (small sample size) has been mentioned as well as the problem that the membrane status of Hsp70 on CECs could not be determined due to lack of material.
The authors want to thank reveiwer 2 for helpful suggestions.
Round 2
Reviewer 1 Report (New Reviewer)
Comments and Suggestions for Authors
The authors have addressed my comments. However, there are still inappropriate expressions and sentences in the Results section. Please revise or delete the phrases below. If necessary, I recommend seeking professional editing services specifically for the Results section.
- After a successful validation~
- Due to the limited availability of primary CECs,
- These data indicate that the origin of CECs isolated via membrane-bound Hsp70 or EpCAM differ.
- This finding shows that circulating cells derived from the blood of endometriosis patients share some typical characteristics with circulating tumor cells (CTCs).
- Genes differentially expressed~
Author Response
Reviewer 1
Comments
The authors have addressed my comments. However, there are still inappropriate expressions and sentences in the Results section. Please revise or delete the phrases below. If necessary, I recommend seeking professional editing services specifically for the Results section.
- After a successful validation~
Response
The sentence has been revised as follows: “The validated cmHsp70.1 mAb- and EpCAM mAb-based approaches for isolating circulating tumor cells (CTCs) from EDTA anti-coagulated blood from patients with different tumor entities [16], was used to isolate ..”
- Due to the limited availability of primary CECs,
Response
The content of this sentence has been shifted to the discussion part
- These data indicate that the origin of CECs isolated via membrane-bound Hsp70 or EpCAM differ.
Response
This sentence has been shifted to the discussion part
- This finding shows that circulating cells derived from the blood of endometriosis patients share some typical characteristics with circulating tumor cells (CTCs).
Response
This sentence has been shifted to the discussion.
- Genes differentially expressed~
Response
“Vice versa” was included in the second sentence beginning with “genes differentially expressed” for clarification.
Response
The English has been revised by the native English speaking co-author Alan Graham Pockley.
The authors want to thank the reviewer for the comments.

Reviewer 2 Report (New Reviewer)
Comments and Suggestions for Authors
- The authors have addressed the comments
Author Response
Comment: The reviewres have addressed my comments.
Response: The quthors want to thank reveiwer 2 for the helpful suggestions.
This manuscript is a resubmission of an earlier submission. The following is a list of the peer review reports and author responses from that submission.
Round 1
Reviewer 1 Report
Comments and Suggestions for Authors
Christian Guder and colleagues studied whether extracellular Hsp70 and circulating endometriotic cells (CECs) reflects the presence and development of endometriosis. One of the key findings is using ELISA, they measured circulating free and vesicle associated Hsp70 levels in patients with endometriosis, pre- and post-surgery and healthy control and their finding shows the mean Hsp70 levels in patients with histologically confirmed endometriosis showed higher level comparing to those of a healthy group.
I have the following questions and suggestions to be addressed.
1/ Why is the measurement of Hsp70 levels among patients with endometriosis showed a large variation regardless of its higher concentration compared to the negative controls?
2/ The error bar in Figure 1 shows there is a big value variation within the same group, and I have a concern that either is a measurement error or calculation error. Strong Justification is required for this.
3/ Table might need statistical analysis to compare the difference between the two groups.
4/ Figure 2 is not clear and the type of microscope brand need to be described. Please put measurement scale bar in different color in the figure. Labelling is poor.
5/ Figures 3 and 4 are not well organized. Please put a, b, and c, in the same panel. Result section is required with brief description before writing figure legend.
6/ It is required to provide flow data in addition to the histogram in figure 4b.
Reviewer 2 Report
Comments and Suggestions for Authors
Please see the attached files, thank you!

No comments in English